# Vibration Characteristics of Corn Combine Harvester with the Time-Varying Mass System under Non-Stationary Random Vibration

**Yanchun Yao** [1,2,3]**, Xiaoke Li** [1,2]**, Zihan Yang** [4,5]**, Liang Li** [1,2]**, Duanyang Geng** [1,2]**, Peng Huang** [1,2]**, Yongsheng Li** [3] **and Zhenghe Song** [4,]*

1   School of Agricultural Engineering and Food Science, Shandong University of Technology, Zibo 255000, China
2   Shandong Provincial Key Laboratory of Dry Farming Machinery and Information, Zibo 255000, China
3   Agricultural Equipment Research Institute of Shandong Wuzheng Group, Rizhao 276825, China
4   Beijing Key Laboratory of Optimized Design for Modern Agricultural Equipment,
    China Agricultural University, Beijing 100083, China
5   Luoyang Smart Agricultural Equipment Institute Co., Ltd., Luoyang 471023, China
*   Correspondence: songzhenghe@cau.edu.cn

**Abstract:** In field harvesting conditions, the non-stationary random vibration characteristics of the harvester are rarely considered, and the results of vibration frequency calculated by different time–frequency transformation methods are different. In this paper, the harvester's vibration characteristics under the time-varying mass were studied, and the correlation between vibration frequency and modal frequency was analyzed. Firstly, under the conditions of time-varying mass (field harvesting conditions) and non-time-varying mass (empty running condition), the non-stationarity characteristics of vibration signals at 16 measurement points of a combined corn harvester frame were studied. Then, fast Fourier transform (FFT), short-time Fourier transform (STFT), and continuous wavelet transform (CWT) were used to calculate the vibration frequency distribution characteristics of the corn harvester. Finally, based on the EFDD (enhanced frequency domain decomposition) algorithm, the correlation between the primary vibration frequency and the operating mode frequency is studied. The results show that the mean, variance, and maximum difference of the vibration amplitude under harvesting conditions (mass time-varying system) are 0.10, 26.5, and 1.0, respectively, at different harvesting periods (0~10 s, 10~20 s, 20~30 s). The harvesting conditions' vibration signals conform to the characteristics of non-stationary randomness. The FFT algorithm is used to obtain more dense vibration frequencies, while the frequencies based on STFT and CWT algorithms are sparse. The correlation between the FFT method and the EFDD algorithm is 0.98, and the correlation between the STFT, CWT, and the EFDD algorithm is 0.99 and 0.98. Therefore, the primary frequency of the STFT methods is closer to the modal frequency. Our research laid the foundation for further study and application of mass time-varying combined harvester system non-stationary random vibration modal frequency identification and vibration control.

**Keywords:** harvester machinery; time-varying mass system; non-stationary random vibration; vibration frequency; modal frequency





## 1. Introduction

When the harvester is working in the field, the harvester's mass is constantly increasing, and the running speed of the harvester keeps changing. The whole machine and essential parts vibrate violently during the harvesting operation [1,2], which affects the physical and mental health of the driver [3], and quickly leads to welding deformation and fracture of the welding structures, such as the harvester's header and frame. Failures frequently occur, which cause seriously decreasing harvesting efficiency, operation accuracy, and harvest losses. Therefore, improving the reliability of the harvester and reducing the

grain loss caused by vibration are crucial ways to increase grain yield and ensure food security [4].

The harvester is thecritical element of the complex soil-machine-crop system. The engine and different vital parts of the harvester (header, peeling device, straw crushing, returning device, etc.) work together and produce complex vibration modes. During the harvesting process, the mass of the whole machine increases with time (the harvested maize enters the grain tank, and the mass of the entire harvester constantly increases).

At present, research on the vibration characteristics of complex systems, such as harvesting machinery, mainly focuses on vibration modeling [5–7], finite element method (FEM) analysis [8–10], and vibration testing [11–13]. In the theoretical modeling aspects, researchers established a 2 or 3 degrees tractor model to deduce the modal frequency, tested the vibration response data, calculated the power spectral density of vibration data to obtain vibration characteristics of the tractors, and verified the accuracy of the models [14–16]. In other methods, based on experimental modal analysis (EMA), the researchers obtained the vibration characteristics of the combined harvester's header and proposed an optimization scheme to improve the dynamic performance of the machine [17]. Chen et al. (2022) [18] used FEM to calculate the modal frequencies of automatic cutting devices. Zhan et al. (2022) [19] built a prediction system to predict the mechanical deformation of a high-speed rice transplanter during the harvesting operation and proposed an efficient vibration characteristics analysis method. Moreover, scholars have analyzed the vibration characteristics of plants using techniques such as FEM and EMA [20,21].

Studies have been conducted on the identification of modal parameters of harvesting machinery, in which the vibration transfer function of the combined harvester cleaning sieves was calculated, providing the modal frequencies of the structures [22,23]. Geng et al. (2021) [24] combined the time–frequency analysis method and power spectral density to obtain the transplanter's vibration characteristics. Based on the EMA method, the modal frequency was identified, and the primary vibration source of the system was determined. Zhou et al. (2018) [25] identified the modal parameters based on the vector–time autocorrelation regression model and the least squares support vector machine method for the linear time-varying system under only the output response. Previous scholars have also researched vibration characteristics and modal parameter identification under operating conditions [26,27]. Raza et al. [28] studied the vibration behavior and operating modal characteristics of the combined harvester header andcorrected the vibration model. Yao et al. (2019) [29] carried out the operating modal test of the corn harvester. Based on the only response vibration data, the SSI and EFDD algorithms were used to identify the modal parameters under harvesting conditions. Reynders [30] used the SSI algorithm to determine the operating modal frequency.

In addition, Ren et al. (2013) [31] tested the vibration transmission feature of the electric multiple units (EMU) and determined the vibration energy transfer path of the structure. Zha et al. (2020) [32] tested the shock and vibration feature of the gearbox bearing and found that the shock acceleration increased with the length of the flat scar. Adam et al. (2020) [33] used the transmissibility method to calculate the vibration frequencies of the tractor seat and explored the suitable seat structure for the human body. Timo et al. (2019) [34] proposeda dynamic energy analysis method to verify the finite element model, calculate the tractor, and demonstrate its vibration feature. Ji et al. (2017). [35] used the confidence criterion method to optimize the modal experiment and optimized the structural parameters of the rice transplanter support arm via the sequential quadratic programming method.

Relevant experts and scholars have carried out a lot of fundamental research on agriculture machinery's stationary random vibration. However, there is less research on the non-stationary random vibration characteristics of the harvester, and the influence of mass–time-varying on the harvester's structure is less considered. This paper studied non-stationary characteristics of random vibrations in field harvesting conditions, used fast Fourier transform (FFT), short-time Fourier transform (STFT), and continuous wavelet

transform (CWT) methods to analyze the corn harvester's vibration frequencies of 16 test points, and compared the results of different analysis methods. We also analyzed the frequency distribution of different time–frequency transform methods, discussed the frequency distribution feature of the mass–time-varying system under non-stationary random vibration conditions, and analyzed the relationship between the primary vibration frequencies and the modal frequency. Our research laid the foundation for further study and application of combined harvester non-stationary random vibration modal frequency identification and vibration control.

## 2. Time–Frequency Analysis

The Fourier transform is developed from the Fourier series. The Fourier transform method can transform the random vibration signals from the time–domain to the frequency domain. The formula [36] is

$$X(\omega) = \int_{-\infty}^{\infty} x(t) e^{-j\omega t} dt$$

where $t$ is the time, $x(t)$ is the time–domain signal, e is the exponential basis, j is an imaginary unit, $\omega$ is the angular frequency, and $X(\omega)$ is the frequency spectrum.

The Fourier transform cannot display the frequency characteristics of a specific period. The short-time Fourier transform method demonstrates both the time and frequency domain attributes. The theoretical basis [37] is

$$G(\omega, b) = \int_{-\infty}^{\infty} x(t) g(t - b) e^{-j\omega t} dt$$

where $t$ is the time, $x(t)$ is the time–domain signal, $\omega$ is the angular frequency, $g(t)$ is the window function, $b$ is the length parameter of the window, and $G(\omega, b)$ is a two-dimensional spectrum.

Mathematically, the continuous wavelet transform uses the wavelet function's inner product to signal function and calculate the degree of similarity between the signal and the wavelet function. Then, the frequency components of the vibration signal are obtained. Its theoretical basis [38] is

$$W_f(a, b) = \frac{1}{\sqrt{a}} \int_{-\infty}^{\infty} x(t) \psi\left(\frac{t - b}{a}\right) dt$$

where $t$ is the time, $x(t)$ is the time–domain signal, $\psi(t)$ is the wavelet basis function, $a$ is the scale parameter (inverse to the frequency), $b$ is the translation parameter (corresponding to time), and $W_f(a, b)$ is a two-dimensional spectrum.

## 3. Operational Modal Test of Corn Harvester Frame

The operational modal test measured the vibration signals of a 4YZP-4HA 4-row corn combine harvester produced by the Wuzheng Group of China [39]. The size parameter of the corn harvester was 6800 mm × 2820 mm × 3400 mm. The mass of the harvester was 7400 kg, and the corn-feeding rate for the harvester was 3.7 kg/s. The harvester was equipped with a 122 kW diesel engine.

The test used a type 3062V 16-channel universal dynamic signal acquisition device to collect the vibration acceleration signals of each measurement point, produced by the China Orient Institute of Vibration & Noise. The single-axis accelerometers by PCB Piezotronics Group were pasted on the frame with glue. The layout of the measurement points is shown in Figure 1.

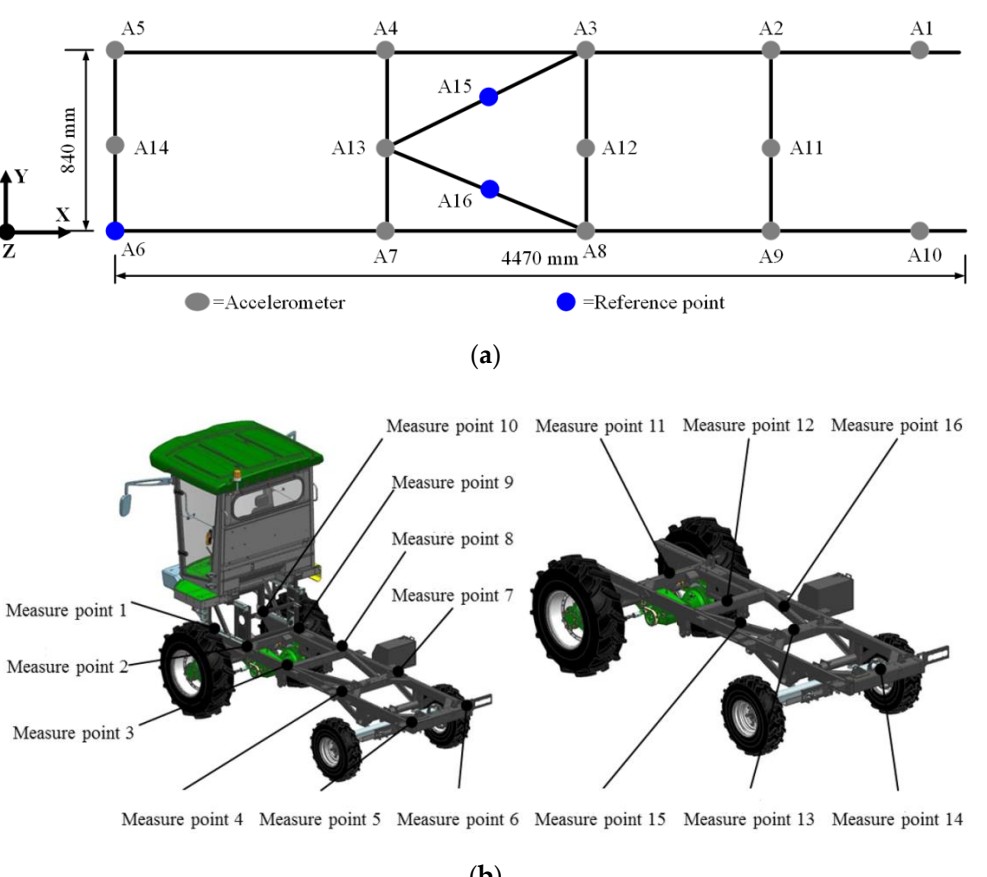

(a)

(b)

**Figure 1.** Measurement points for the harvester frame modal test: (**a**) the layout of measurement points, (**b**) the installation positions of accelerometers on frame assembly.

The device installation positions of the operational modal test on the harvester frame are shown in Figure 2.

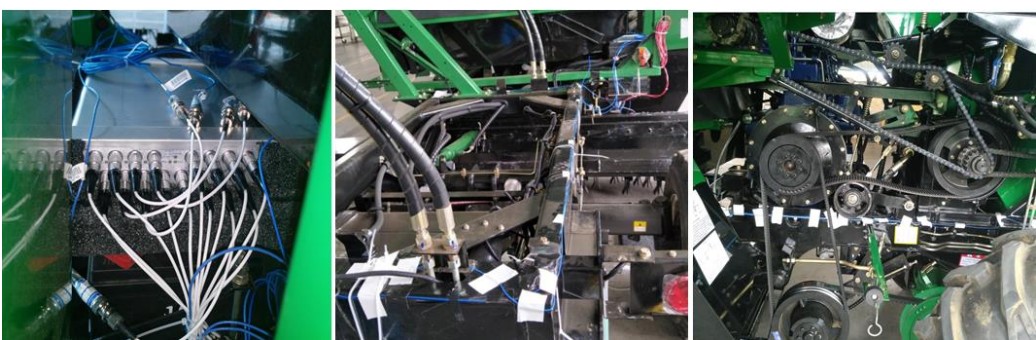

**Figure 2.** Operating modal test of the harvester frame.

The test location was Qingdao, China. In the test, the corn variety was Chenghai 605, and the corn was in the mature period, the water content of the grain was 22.8%, the spacing of corn rows was 650 mm, the spacing of plants was 250 mm, and the minimum maize height was 790 mm, which met the requirement of corn-harvesting conditions. The soil was yellow clay, the soil moisture content was 24.6%, the soil compatibility was 129.2 kPa, the test environment temperature was 17.4~19.2 °C, and the humidity was 37.6~40.9%; the wind force was level 2, and the wind direction was southeast. The operational modal test pictures are shown in Figure 3.

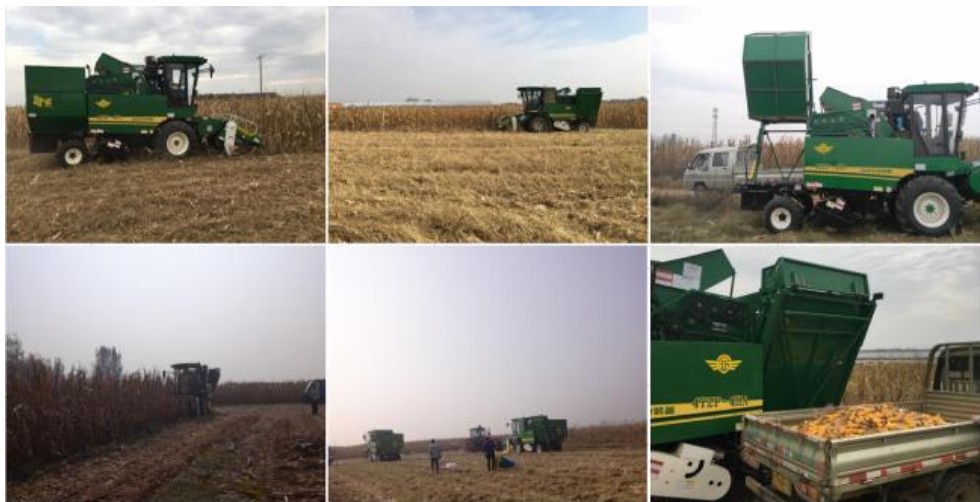

**Figure 3.** Operational modal test of harvester frame in the field.

## 4. Time and Frequency Domain Characteristics of Harvester's Vibration Signals

*4.1. Time Domain Characteristics of Vibration Signals*

Generally, mean value, variance, and root mean square (RMS) value are used to determine the stationarity of signals. The time–domain characteristic of stationary random signals in different periods is a fixed constant. In contrast, the time–domain feature of non-stationary random signals in different periods has significant differences [40,41].

In running conditions (without harvest) and harvesting conditions, the corn harvester was excited bythe engine, working parts, field, and complex vibration modal. Therefore, it is necessary to study the stationary characteristics of vibration signals of harvesting machinery. The time–domain characteristics of three different periods (0~10 s, 10~20 s, and 20~30 s) are shown in Figure 4.

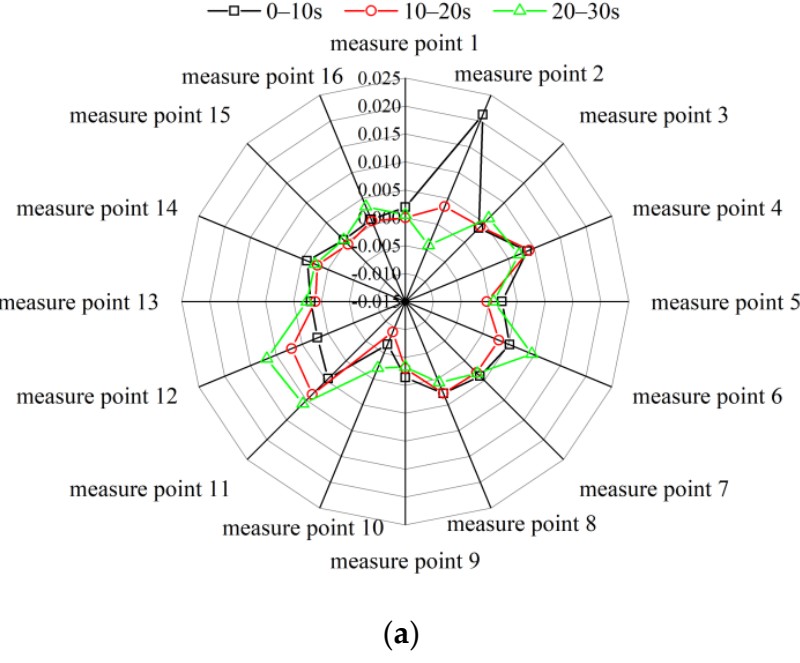

(a)

**Figure 4.** *Cont.*

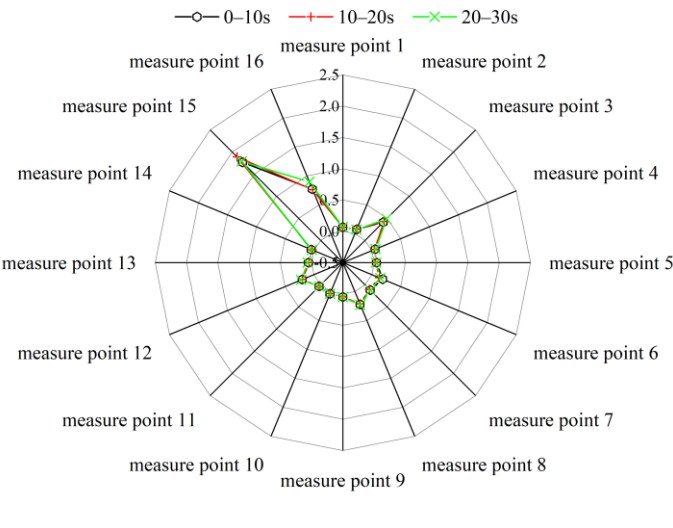

(**b**)

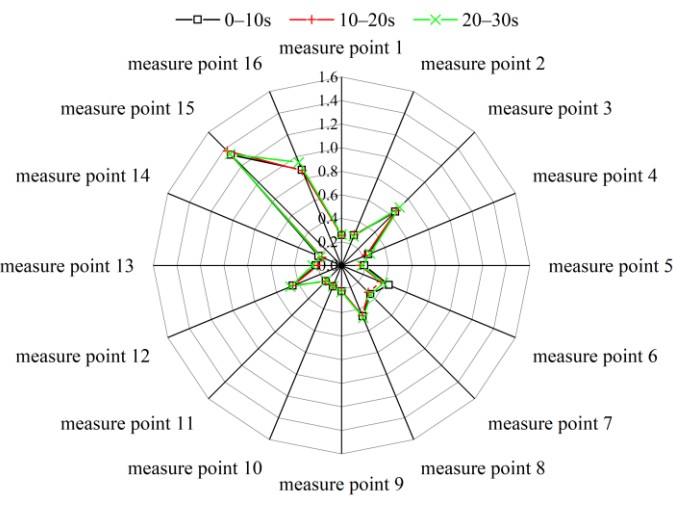

(**c**)

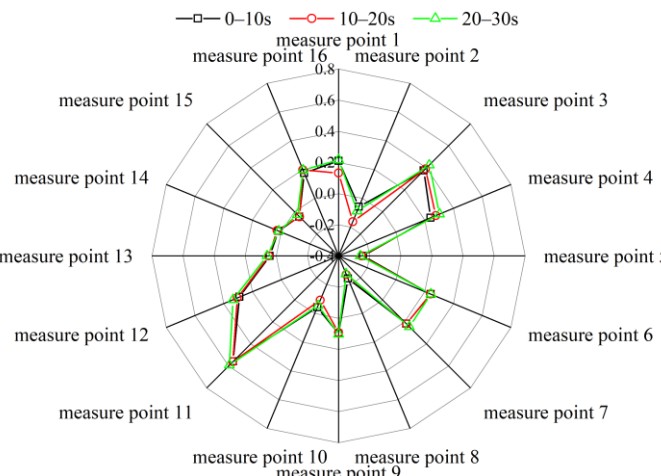

(**d**)

**Figure 4.** *Cont.*

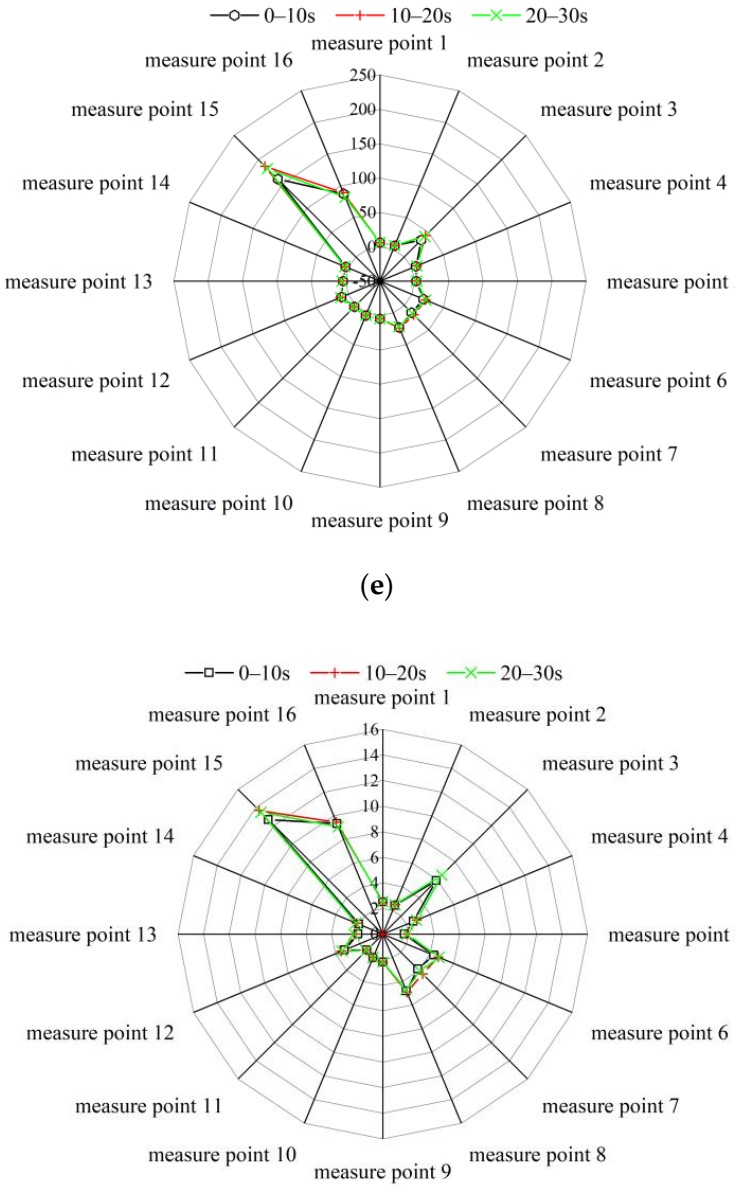

(**e**)

(**f**)

**Figure 4.** Comparison of time–domain characteristics under different conditions: (**a**) mean values of running condition, (**b**) variance of running condition, (**c**) RMS values of running condition, (**d**) mean values of harvesting conditions, (**e**) variance of harvesting conditions, (**f**) RMS values of harvesting conditions.

It can be seen from Figure 4 that the statistical characteristics of time–domain vibration signals significantly differ in running and harvesting conditions. The mean value, variance, and RMS value are small in running conditions. The maximum difference between the mean, variance, and RMS values of the three periods are 0.02, 0.14, and 0.08, respectively. The time–domain characteristics are stable, and it can be considered that the vibration signals in running conditions are approximately stationary.

In harvesting conditions, the difference in mean values, variance, and RMS values of the vibration signals are more significant than in the running states. The maximum difference between the mean, variance, and RMS values of the three time periods are 0.10, 26.5, and 1.0, respectively. The maximum difference between the variance and the RMS values appears at measurement point 15. In addition, the difference between other

measurement points is also more significant than the running condition. It can be seen that the vibration signals of the corn harvester are non-stationary and random under harvesting conditions.

### 4.2. Frequency Domain Characteristics of Non-Stationary Random Vibration Signals

Three methods (FFT, STFT, and CWT) are used to analyze the frequency distribution of non-stationary vibration.

#### 4.2.1. Vibration Frequency Distribution Characteristics of FFT

In the vibration signal sample system, the sampling frequency was 500 Hz, the analysis frequency was 250 Hz, and the number of spectral lines was 112,500. Since the peaks of the spectrum reflect the contribution of the vibration frequency components used in MATLAB to write the FFT program, the vibration frequencies corresponding to each peak in the spectrum were extracted, and the vibration frequencydistribution of 16 measurement pointswas obtained, as shown in Figure 5.

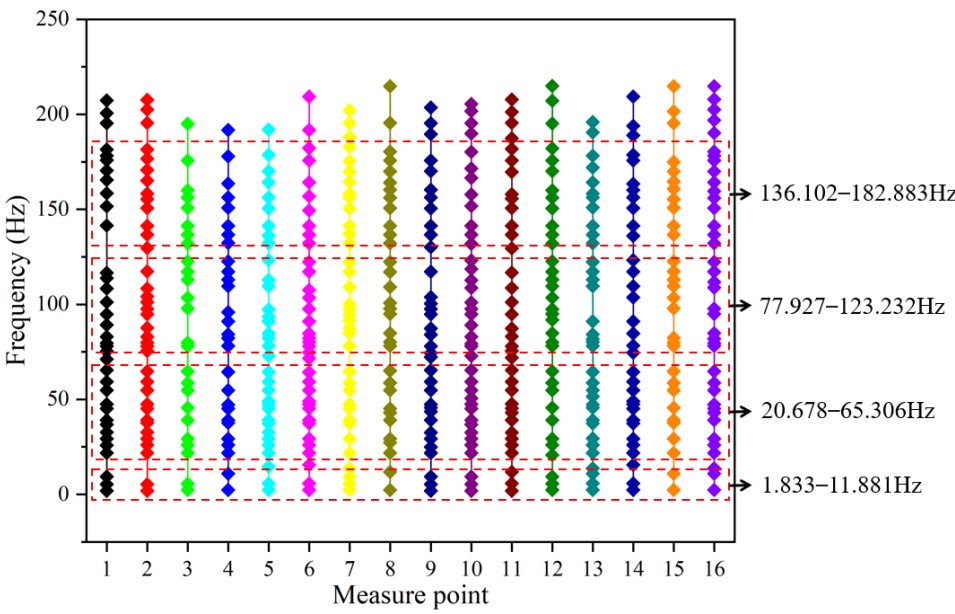

**Figure 5.** Vibration frequency distribution in harvesting conditions (FFT).

It can be seen from Figure 5 that the vibration frequencies of the harvesting conditions are mainly concentrated in 1.8~11.9 Hz, 20.7~65.3 Hz, 77.9~123 Hz, and 136~183 Hz. The frequency distribution obtained by the FFT method is relatively dense, and there will be multiple peaks in a small frequency range, which brings difficulties in determining the main vibration frequencies of the system.

#### 4.2.2. Vibration Frequency Distribution Characteristics of STFT

We wrote a MATLAB program to analyze the vibration signals of the corn harvester using the STFT method. In the analysis, the selected Hamming window was 1024 lengths to truncate the original signals, and the overlap of the truncation was half of the window length. The vibration frequencies with the largest amplitude in the whole time–domain were extracted, and the distribution of vibration frequencies at different measurement points is shown in Figure 6.

The vibration frequency distribution in Figure 6 obtained by the STFT method is mainly concentrated in 2.0~5.8 Hz, 21.0~64.9 Hz, 78.1~124 Hz, and 130~167 Hz. The frequency components obtained by STFT are less than FFT analysis results, especially in 78.1 Hz to 124 Hz. STFT has a low resolution in a low-frequency range, and only two vibration frequencies were observed.

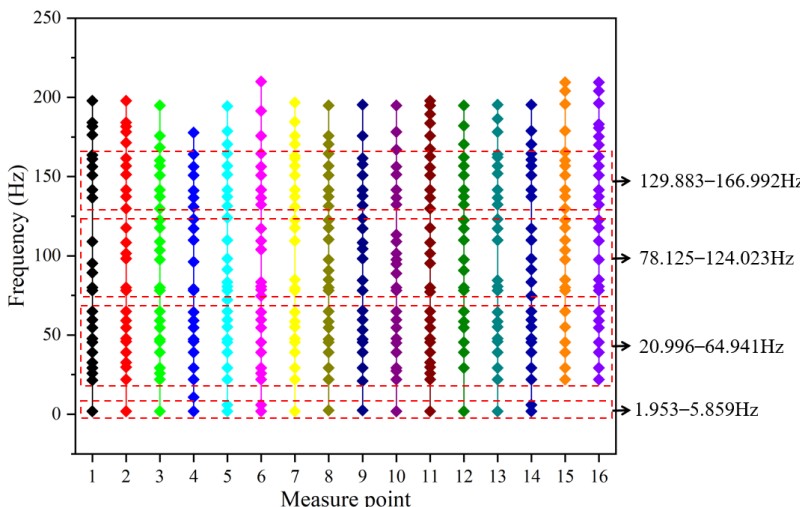

**Figure 6.** Vibration frequency distribution in harvesting conditions (STFT).

### 4.2.3. Vibration Frequency Distribution Characteristics of CWT

We wrote a CWT program and selected the "Cmor3-3" wavelet to analyze non-stationary random vibration signals. The vibration frequencies with the largest amplitude in the time–domain were extracted from the spectrum, and the distribution of vibration frequencies at different measurement points is shown in Figure 7.

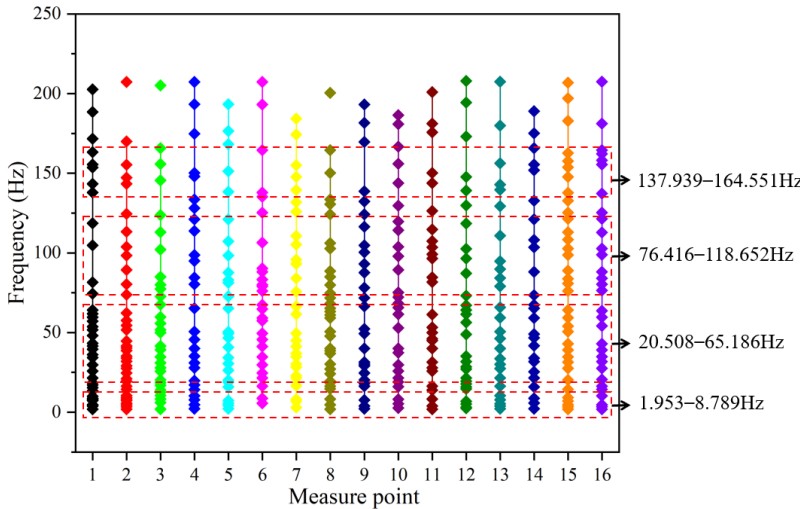

**Figure 7.** Vibration frequency distribution in harvesting conditions (CWT).

The vibration frequencies shown in Figure 7, calculated by the CWT method, are mainly concentrated in 2.0~8.8 Hz, 20.5~65.2 Hz, 76.4~119 Hz, and 138~165 Hz. Compared with FFT and STFT, CWT identified more frequency components in the low-frequency range (2~15 Hz) than the STFT algorithm, but fewer than the FFT algorithm. In addition, the CWT algorithm identified fewer frequency components in 80~220 Hz.

It can be seen that the frequencies obtained by the FFT method are relatively dense, and that most of the measurement points have more than 35 vibration frequency peaks. The frequencies obtained by the STFT and CWT methods are relatively sparse, and the frequency peaks of different measurement points are mostly less than 30.

### 4.2.4. Comparison of Vibration Frequency Distribution Characteristics

The vibration frequency spectra of the corn harvester calculated by the three methods show that most of the vibration frequency peaks appear at 22 Hz, 26 Hz, 29 Hz, 33 Hz, 39 Hz, 46 Hz, and 48 Hz, and their multipliers.

In Figure 8, the frequency spectra of the corn harvester under harvesting conditions obtained by FFT, STFT, and CWT are relatively similar. The time–frequency spectra were obtained by STFT and CWT methods at 0~10 Hz. The frequency resolution of the STFT method in 0–10 Hz is low, mainly due to the feature of the window function. The vibration frequency peaks of the STFT method are distributed continuously in the time domain. The frequency resolutionabove 150 Hz of the CWT spectrum is low, and the resolution of the CWT spectrum in the time domain is higher.

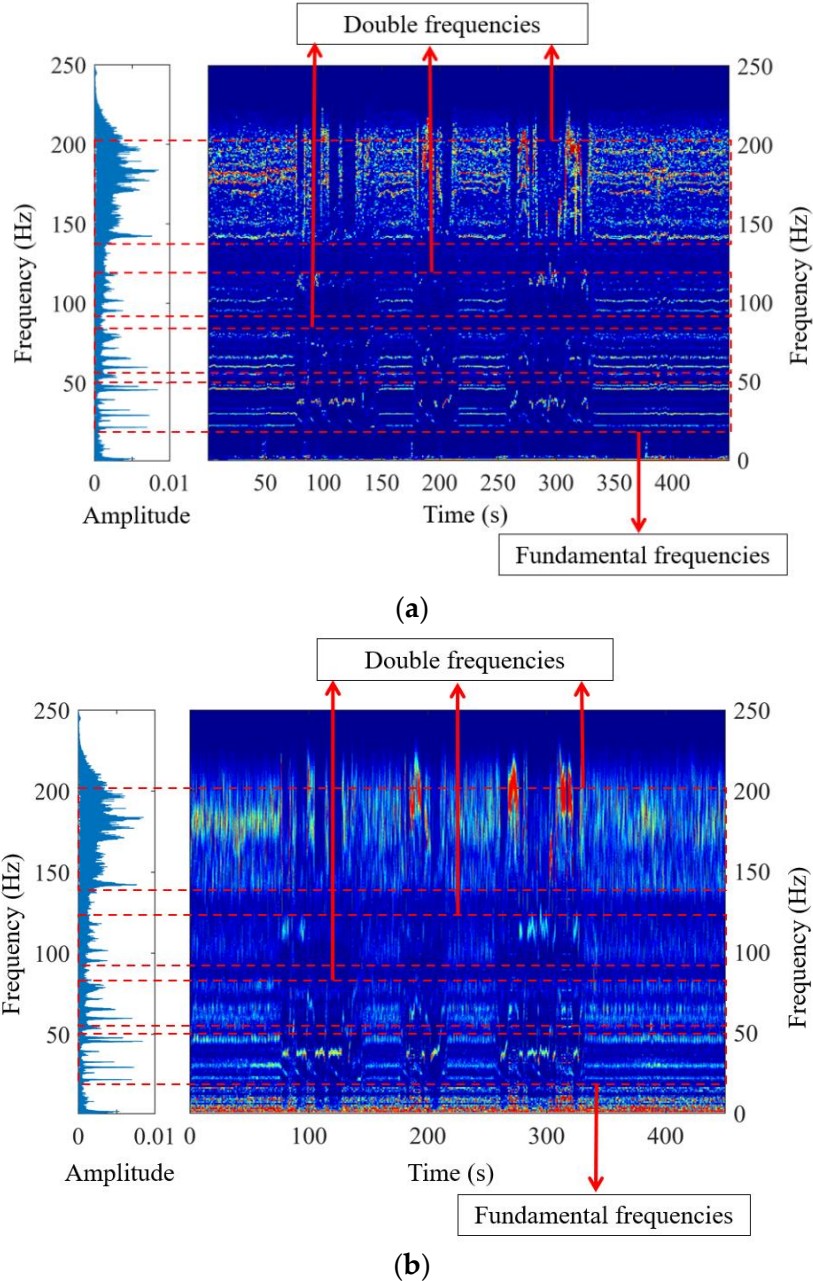

**Figure 8.** Comparison of spectra: (**a**) comparison of FFT and STFT spectra, (**b**) comparison of FFT and CWT spectra.

FFT cannot display the frequency-changing detailswith the mass of vibration signals using STFT and CWT time–frequency analysis methods. The frequency distribution feature of the harvester at different masses obtained by STFT and CWT time–frequency analysis methods is shown in Figure 9.

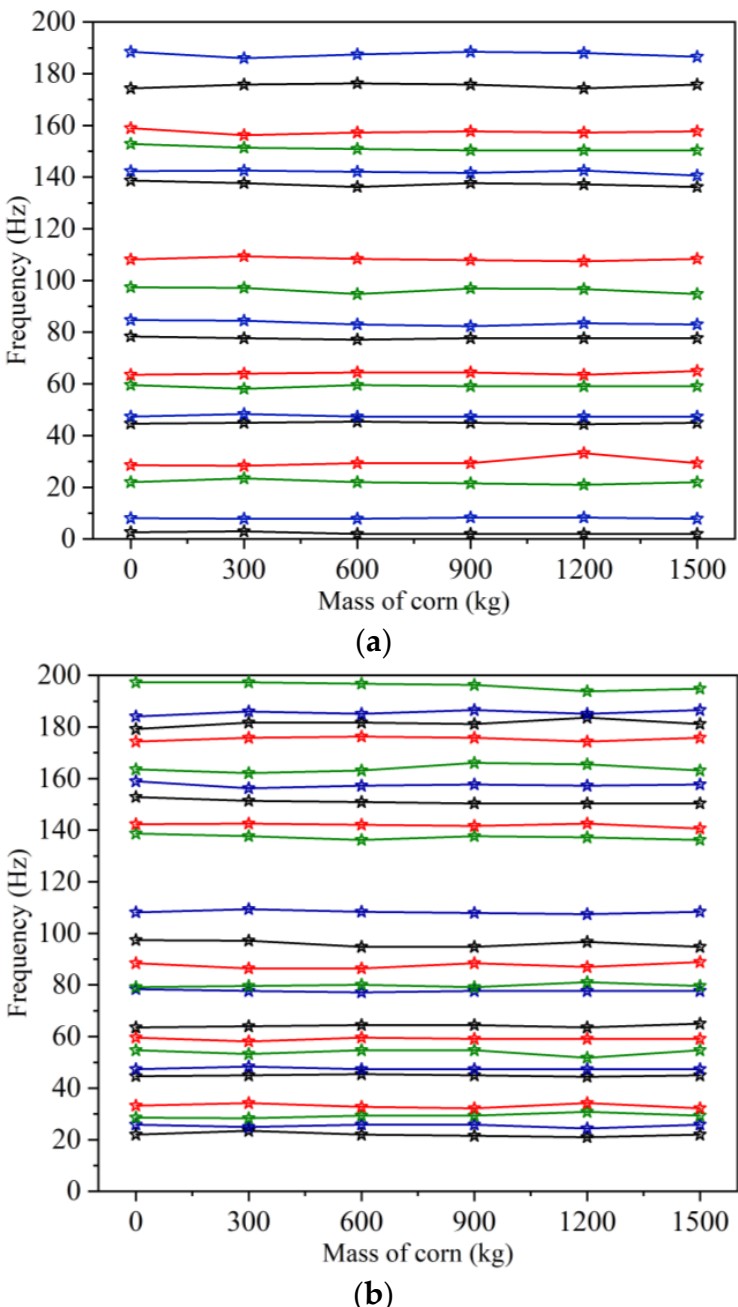

**Figure 9.** Frequency distribution ofmass time-varying system: (**a**) frequency distribution under time-varying mass condition based on STFT method, (**b**) frequency distribution under time-varying mass condition based on CWT method.

In harvesting conditions, the feeding amount of corn was 3.7 kg/s, and the mass of the whole machine increased approximately linearly with time. In Figure 9, the vibration frequency peaks correspond to different masses arranged in approximately straight lines, and the frequencies extracted from STFT and CWT spectra vary little with mass. The difference between the frequencies corresponding to each point in the same line is mostly less than 1 Hz, the maximum difference between STFT is 3.4 Hz, and the maximum difference between CWT is 4.9 Hz. The nonlinear relationship between vibration frequencies and mass is not apparent. It can be considered that when the feeding amount is small, the mass change has a relatively small influence onthe harvester's vibration frequencies.

## 5. Correlation of Vibration Frequencies and Modal Frequencies

It is difficult to obtain the precise excitation function in harvesting conditions. Therefore, based on only the response signals of 16 measurement points, the enhanced frequency domain decomposition (EFDD) algorithm was used to identify the modal frequencies of the corn harvester frame under harvesting conditions. The 18th-order modal frequencies are shown in Table 1.

**Table 1.** Modal values of EFDD algorithm in harvesting conditions.

| Orders | Enhanced Frequency Domain Decomposition (EFDD) | |
| --- | --- | --- |
| | Modal Frequency (Hz) | Damp Ratio (%) |
| 1 | 21.868 | 1.111 |
| 2 | 29.300 | 0.410 |
| 3 | 39.151 | 0.568 |
| 4 | 45.409 | 0.804 |
| 5 | 54.627 | 0.516 |
| 6 | 59.241 | 0.996 |
| 7 | 64.934 | 0.881 |
| 8 | 77.970 | 0.446 |
| 9 | 79.786 | 0.574 |
| 10 | 84.618 | 0.880 |
| 11 | 97.723 | 0.609 |
| 12 | 109.371 | 0.632 |
| 13 | 117.189 | 0.458 |
| 14 | 122.542 | 0.493 |
| 15 | 136.669 | 0.460 |
| 16 | 141.625 | 0.441 |
| 17 | 150.614 | 0.471 |
| 18 | 156.156 | 0.447 |

### 5.1. Comparison of Primary Frequencies and Modal Frequencies under Harvesting Conditions

The 18 primary vibration frequencies, which corresponded to the highest amplitude and were distributed in the time–domain continuously, were extracted from the FFT, the STFT, and the CWT spectrum. Comparing 18 main frequencies with the modal frequencies of the EFDD algorithm, the result is shown in Figure 10.

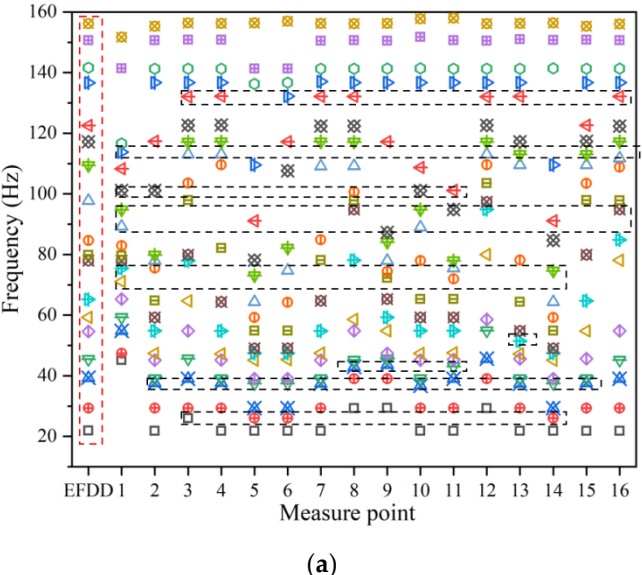

(**a**)

**Figure 10.** *Cont.*

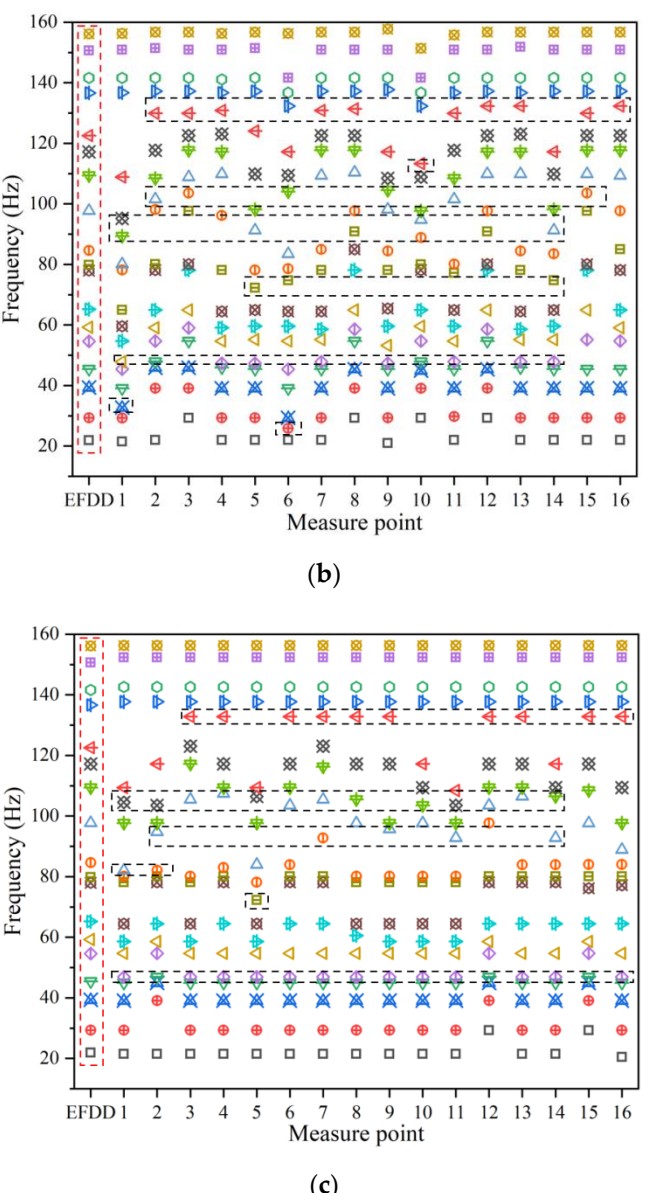

**Figure 10.** Comparison of primary frequencies and EFDD modal frequencies: (**a**) Comparison of FFT-primary frequencies and EFDD modal frequencies, (**b**) Comparison of STFTprimary frequencies and EFDD modal frequencies, (**c**) Comparison of CWTprimary frequencies and EFDD modal frequencies.

In Figure 10, the first column is modal frequencies identified by the EFDD algorithm (shown as the red line in Figure 10a–c), while other columns are 18 primary frequencies (corresponding to 18 highest amplitude) extracted from the FFT, STFT and CWT spectrum of 16 measurement points. Among the 18 primary frequencies of different measurement points, nine frequencies range of the FFT method (indicated by the black line in Figure 10a) have large deviations with modal frequencies. Meanwhile, STFT and CWT algorithms appear eight frequency ranges (shown by the black line in Figure 10b,c) and significant variations with modal frequencies. These vibration frequencies cannot be identified as modal frequencies.

*5.2. Correlation of Primary Frequencies and Modal Frequencies under Harvesting Condition*

This paper analyzed the correlation between the primary vibration frequencies of the 16 measurement points extracted from the FFT, the STFT, and the CWT methods and the 18-order modal frequencies identified by the EFDD algorithm, respectively. The results

are shown in Figure 11. The average correlation between the main vibration frequencies extracted by the FFT method of 16 measurement points and the modal frequencies identified by EFDD is 0.98, and the moderate correlation between the STFT, CWT methods, and the modal frequencies are 0.99 and 0.98, respectively. The frequency obtained from the STFT method is closer to modal frequencies identified by EFDD algorithms. The correlation between the vibration frequencies calculated by STFT algorithms and the modal frequencies is better.

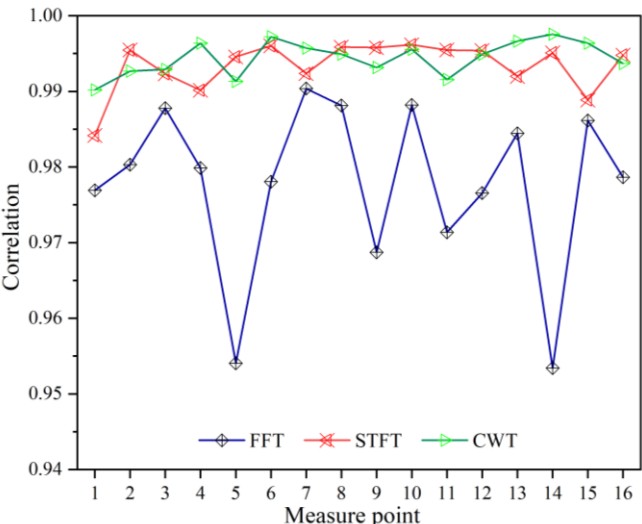

**Figure 11.** Correlation of vibration frequencies calculated by different methods and modal frequencies.

## 6. Conclusions

This study aims to find a more suitable method to analyze the vibration characteristics of the harvester under complex working conditions and provide an idea to identify the modal frequency.

(1) It was found that the random vibration characteristics of the corn harvester are approximately stationary in running conditions (non-time-varying mass system). The vibration signals in the harvesting operation conform to non-stationary random vibration characteristics (time-varying mass system).

(2) The vibration frequencies of the corn harvester calculated by FFT were relatively dense, and most measurement points had more than 35 frequency peaks in the frequency spectrum. The vibration frequencies obtained by STFT and CWT methods were relatively sparse, with fewer than 30 frequency peaks at the most measurement point.

(3) Under harvesting (time-varying mass) conditions, the increased mass had a negligible effect on vibration frequencies, and the nonlinear relationship between vibration frequencies and mass was not apparent.

(4) Under the non-stationary random characteristics condition, the average correlation degree of the main vibration frequencies obtained by the FFT, STFT, and CWT methods and the modal frequencies were 0.98, 0.99, and 0.98, respectively. The primary frequencies of the STFT method were more likely to correspond with operating modal frequencies. The STFT method was more suitable for analyzing the harvester's signal.

The study provides a new idea for identifying modal frequencies from non-stationary random vibrations in complex harvesting environments. Furthermore, it will provide a reference for vibration control, dynamic design, and fault diagnosis of agricultural machinery.

**Author Contributions:** Conceived the proposed idea, Y.Y., X.L. and Z.S.; designed experiment, Y.Y., Z.S., Z.Y. and Y.L.; analyzed the data of investigation, X.L. and P.H.; writing—original draft preparation, Y.Y., X.L. and L.L.; writing—review and editing, Y.Y. and D.G. All authors have read and agreed to the published version of the manuscript.

**Funding:** All authors would like to acknowledge the financial support from the National Key R&D Program of China (2021YFD2000502) and China Postdoctoral Science Foundation (2022M711982).

**Institutional Review Board Statement:** Not applicable.

**Informed Consent Statement:** Not applicable.

**Data Availability Statement:** Not applicable.

**Conflicts of Interest:** The authors declare that they have no known competing financial interests or personal relationships that could have appeared to influence the work reported in this paper.

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
