# Peer review of "Vibration Characteristics of Corn Combine Harvester with the Time-Varying Mass System under Non-Stationary Random Vibration"

_agriculture, doi:10.3390/agriculture12111963_

Round 1

Reviewer 1 Report

This paper studies the vibration characteristics of the corn harvesting machinery under a time-varying mass system, in the time domain, corn harvesting machinery's non-stationary random vibration characteristics are analyzed. In the frequency domain, the time-frequency methods of Fourier transform, short-time Fourier transform and wavelet transform are used to analyze the vibration time-domain signals, and the vibration frequency distribution characteristics under different methods are obtained.

The frequency characteristics of the whole corn combined harvester in the mass change process are obtained. Based on the response data of the harvester under the operating state and the modal operating results, the correlation between different analysis methods and modal frequencies is compared. The structure of the full text is compact, and the analysis is relatively comprehensive. The research has a good reference value for the vibration characteristics analysis of agricultural machinery under harvesting.

However, there are still some details to be improved:

1. It is recommended to provide detailed installation information on the vibration sensor.

2. In the result and discussion part, could you consider a more concise and clear logical explanation? It is suggested to use simple sentences to describe and analyze directly.

3. Improve the English language, check the variables and formulas in the full text, whether the variables have been explained one by one.

Reviewer 2 Report

The comments and suggestions are given in the attached file
